# Study on Spatiotemporal Evolution and Driving Forces of Habitat Quality in the Basin along the Yangtze River in Anhui Province Based on InVEST Model

Yong Cao [1,2], Cheng Wang [1,*], Yue Su [1], Houlang Duan [3], Xumei Wu [1], Rui Lu [1], Qiang Su [1], Yutong Wu [1] and Zhaojun Chu [1]

1 School of Economics and Management, Anhui Agricultural University, Hefei 230036, China; coyg1982@163.com (Y.C.)
2 Anhui Real Estate Registration Center (Anhui Provincial Land and Resources Reserve Development Center), Hefei 230091, China
3 Key Laboratory of Ecosystem Network Observation and Modelling, Institute of Geographic Sciences and Natural Resources Research, Chinese Academy of Sciences, Beijing 100101, China
* Correspondence: wangcheng1991@ahau.edu.cn

**Abstract:** The basin along the Yangtze River in Anhui Province is an important ecological protective screen in the Yangtze River Delta Economic Belt, with a large number of wetlands, lakes, and nature reserves in the basin. The effect of the rapid development of regional urbanization on the ecological environment quality has become an important threat source that restricts ecosystem function and biodiversity protection in the basin. Therefore, this study used InVEST model to analyze the spatial and temporal evolution of habitat quality based on remote sensing image data from 1990, 2000, 2010, and 2020 in the basin along the Yangtze River in Anhui Province and revealed the spatial evolution trend of habitat quality degradation by using hot and cold spot analysis methods. The geographical detector model was used to discuss the main driving factors of habitat quality change. The study results showed that a trend of increase and decrease of construction land and paddy land in the basin from 1990 to 2020 was the opposite, that is, the area of construction land increased, and the area of paddy land decreased. Especially, the area of construction land increased from 390.18 km$^2$ in 1990 to 1616.34 km$^2$ in 2020. The area of increase and decrease of other land types remained around 1% to 2%. During the period from 2000 to 2020, Construction land was mainly transferred in from paddy land, accounting for over 60% of the area transferred in, which indicated the continuous increase of human activity intensity in the study area. From 1990 to 2020, the areas with a significant decline in habitat quality in the basin were mainly concentrated along the Yangtze River and in the northern part of the Chaohu Lake. The area proportion with the lowest grade of habitat quality showed a trend of increasing year by year, that is, the area proportion increased from 4.85% in 1990 to 8.47% in 2020. The hot spots of habitat quality degradation were concentrated in Hefei and its surrounding areas, while the cold spots of the degradation were mainly concentrated in the southern and western mountainous areas. Land use type was the main driving factor affecting habitat quality, and the interaction between land use and per capita GDP was the main driving factor for changes in habitat quality. The study results had important theoretical and practical value for the ecological environment protection and harmonious development of the relationship between humans and nature in the basin along the Yangtze River in Anhui Province.

**Keywords:** basin; habitat quality; InVEST model; hot and cold spot analysis; geographical detector

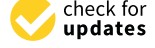



## 1. Introduction

In recent years, biodiversity conservation has gradually become one of the most important aspects of ecological environmental governance and the implementation of ecological environmental protection strategies in various countries around the world [1].

With the continuous development of global population and economy, the role of biodiversity in promoting human well-being has been gradually emerging, and it has gradually become one of the important indicators to measure the ecological environment governance and ecological civilization of some regions [2,3]. However, in recent years, with the continuous expansion of human activities' interference with species' habitats, the decline in global biodiversity has become increasingly severe [4,5]. Habitat quality measured the quality of life and ecological conditions of species within an ecosystem to some extent, which reflected the effect of regional ecological environment changes on biodiversity [6]. Therefore, the implementation of habitat quality assessment would help to reveal the changes in regional biodiversity levels and would have important practical significance for maintaining regional ecological security patterns and formulating policies for the sustainable development of humans and nature.

There were currently two main forms of research related to habitat quality assessment. One form was to obtain the environmental factor data required for the assessment through a large number of field investigations, and on this basis, these researchers constructed a habitat quality assessment index system and used mathematical models to calculate habitat quality [7,8]. This method required high integrity of field data and was only suitable for small size scales. It was difficult to obtain detailed field survey data resources for regional-scale research [9]. Another form was the quantitative assessment of habitat quality using ecological models, which has become the mainstream means of ecosystem assessment in recent decades [10–12]. Currently, ecological models were widely used including InVEST model [11], HIS model [13], MAXENT model [14], etc. These models were based on computer algorithms and achieved a visual display of habitat quality through pre-input and adjustment of ecological parameters, which promoted the development of biodiversity conservation research. It should be pointed out that among numerous ecological models, the InVEST model is widely used in the quantitative spatial assessment of ecosystem services [15], and this model mainly evaluates habitat quality based on the comprehensive impact of various threats on the substrate where the habitat patches are located. Therefore, it has a certain comprehensive advantage in regional habitat quality assessment. However, other models heavily rely on the support of sample points or species distribution data, which would easily lead to bias in evaluation results, and these models weaken the impact of differences in habitat patches or landscape structure types. From the perspective of existing research content, current research on habitat quality mainly focused on the following two aspects. One aspect was the assessment of habitat quality in the main living areas of animal and plant communities [16,17]. This aspect mainly explored the temporal and spatial impacts of human activities, climate, and environmental changes on the quality of species' habitats [16,18]. The second aspect was to assess the spatial and temporal evolution of overall habitat quality under regional human activities and land use changes by conducting regional or basin habitat quality assessments on a regional scale [19,20]. It could be seen from the above content that with the widespread application of ecological models and the gradual improvement of global land cover data, the relevant land cover data was used to carry out spatiotemporal evolution research on regional and even global habitat quality has gradually rising in recent years [21], which also provided reliable technical support for the smooth development of this study.

Although habitat quality assessment has been widely used at basin scales [22,23], there was still a gap in the study of habitat quality in the basin along the Yangtze River in Anhui Province (BAYRAP) [24]. As an important ecological protective screen in the middle and lower reaches of the Yangtze River, the BAYRAP is a key area for biodiversity conservation in the Yangtze River Delta Economic Belt and plays a very important role in maintaining biodiversity and ecosystem stability in China. Under the influence of macro policies such as Anhui's integration into the Yangtze River Delta Economic Belt, human activities, and development intensity in the BAYRAP has continued to expand in the past 30 years, and the contradiction between economic development and ecological conservation has gradually become prominent. The western and southern mountainous areas within the basin are

typical ecologically fragile zones. Many lakes and wetlands around the mainstream and tributaries of the Yangtze River are wintering grounds and habitats for key rare and endangered species. The assessment of the basin habitat quality had important reference value for clarifying the current situation of biodiversity conservation and the shortcomings in the construction of ecological civilization in the BAYRAP and was also an important link in implementing the national strategy of "to step up conservation of the Yangtze River and stop its over-development" in the Yangtze River basin.

Therefore, this study used InVEST model and the spatial analysis methods of ArcGIS software to reveal the spatiotemporal evolution of habitat quality in the BAYRAP based on four issues of remote sensing image data from 1990 to 2020 and explored the driving forces of natural and socio-economic conditions on the evolution of habitat quality by using geographical detector models. This study quantified the spatiotemporal evolution of habitat quality at the basin scale, providing a decision-making basis and data support for macro regulation of land use, regional biodiversity conservation, and ecological protective screen construction.

## 2. Materials and Methods

### 2.1. Study Area

The basin along the Yangtze River in Anhui Province (BAYRAP) is located at 116°35′ E~119°21′ E and 29°31′ N~33°06′ N (Figure 1). It is the main distribution area of the north-south tributaries of the Anhui section of the Yangtze River. The basin covers 57 counties in eight cities, including Anqing City, Chizhou City, Tongling City, Wuhu City, Maanshan City, Xuancheng City, Chuzhou City, and Hefei City in Anhui Province, with a total basin area of over 100,000 km$^2$. The BAYRAP is not only an important component of the Yangtze River Delta Economic Belt but also the core area for the construction of ecological protective screens in the middle and lower reaches of the Yangtze River. The basin has a unique historical and cultural landscape and natural geographical environment and is rich in natural resources, including many resources such as waterpower, minerals, forests, wetlands, biology, and tourism. This basin has an International Important Wetland of Shengjin Lake and more than ten various other national and provincial nature reserves. Lakes and wetlands dotted along the Yangtze River. They are important wintering and breeding grounds for dozens of key endangered wild animals such as white-headed cranes (*Grus monacha*) and Chinese alligators (*Alligator sinensis*) in China [25,26]. Therefore, the BAYRAP has become the region with the richest biodiversity in the Yangtze River Delta region and has been also the main position for ecological civilization and biodiversity conservation in the Yangtze River Delta Economic belt. In recent years, with the rapid development of population and urban scale in the BAYRAP, the contradiction between economic and social development and biodiversity conservation in the region has gradually become prominent [27,28]. Therefore, timely assessment of basin habitat quality has been of great significance for basin ecological environment conservation and biodiversity improvement.

### 2.2. Data Source and Processing

This study involved the land use data of BAYRAP in 1990, 2000, 2010, and 2020, which were from the Resource and Environmental Science and Data Center of the Chinese Academy of Sciences (https://www.resdc.cn, accessed on 8 December 2020). The spatial resolution of the data was 30 m. According to the system of "Classification of Land Use Status" (GB/T21010-2017), the land use data was divided into 17 secondary categories, and the data coordinate system was uniformly converted to WGS_84_Albert.

Previous studies have shown that changes in natural and socio-economic conditions have varying degrees of effect on regional habitat quality [29,30]. In particular, regional water resources, climate, and soil conditions had a significant effect on the shaping of species' habitats. The good or bad natural conditions restricted the level of regional biodiversity, while human production and life had a more profound effect on the regional ecological environment. Therefore, on the basis of referring to existing research results [29–33], this

study, based on the ecological environment status of the BAYRAP, followed the principles of scientificity and accessibility, set twelve factors as independent variables from both the natural environment and socio-economic aspects to explore the driving forces for the spatiotemporal evolution of habitat quality in the basin.

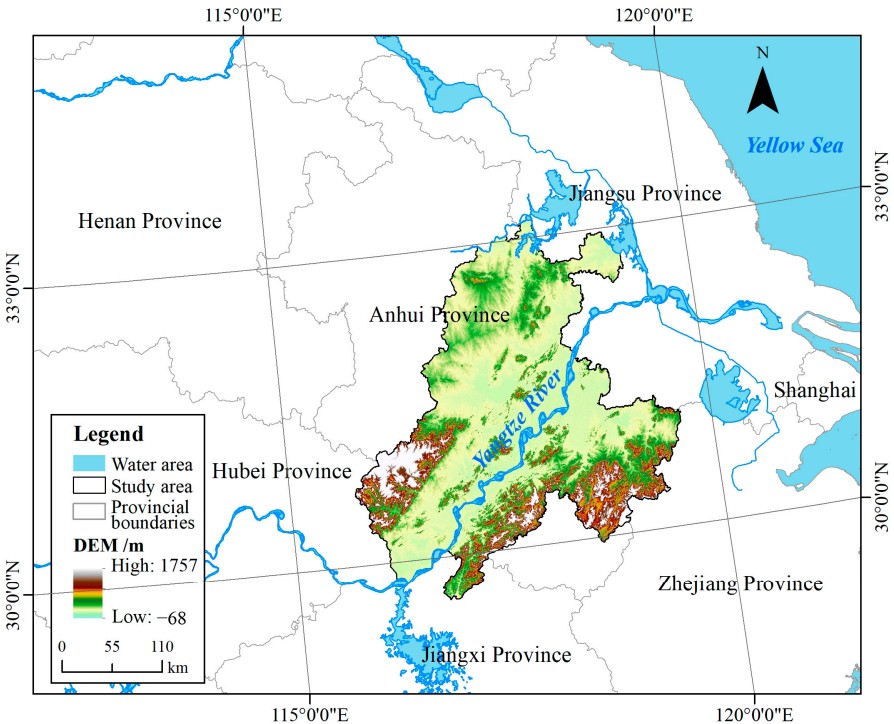

**Figure 1.** Location map of the study area.

The BAYRAP belongs to a subtropical monsoon climate zone, with significant intra and inter-annual changes in temperature and precipitation, which reflected differences in the living environment conditions of species. The basin is a transitional zone from mountainous hills to coastal plains, with significant topographic fluctuations, resulting in significant changes in elevation and slope within the basin. NDVI, as an indicator of vegetation ecological status, reflects the level of vegetation health in the basin. Water network density and soil type reflect the water and soil conditions within the basin. In socioeconomic conditions, GDP, population, roads, natural reserves, and land types all bear the breadth and depth of human activities, which reflected the potential effect of human production and life on the regional ecological environment and ecosystem structure. The data sources of the above indicator system were described in Table 1.

### 2.3. Methodology

#### 2.3.1. InVEST Model and Habitat Quality Evaluation Process

The InVEST model simulated the changes in the quality and value of ecosystem services under different land cover scenarios. It provided the scientific basis for decision-makers to weigh the benefits and impacts of human activities, which realized the spatialization of quantitative evaluation of ecosystem service function value [34]. The Habitat Quality Model in the InVEST model has been widely used in regional or basin habitat quality evaluation in recent years and has achieved rich research findings [31,35,36]. To run the habitat quality module, five necessary data should be input, namely: land use/land cover data, threat factor table, threat source layer data, habitat type and sensitivity table of habitat type to threat factors, and half-saturation coefficient.

**Table 1.** Indicator system for detecting factors affecting habitat quality.

| Dependent Variable | Influence Factor | Independent Variables | Data Source/Description |
|---|---|---|---|
| Habitat quality | Natural environment | Annual precipitation ($X_1$) | China National Meteorological Science Data Center (http://data.cma.cn/ accessed on 12 December 2020) |
| | | Annual temperature ($X_2$) | China National Meteorological Science Data Center (http://data.cma.cn/ accessed on 12 December 2020) |
| | | Altitude ($X_3$) | National Aeronautics and Space Administration |
| | | Slope ($X_4$) | Obtained by DEM calculation |
| | | NDVI ($X_5$) | Resource and Environmental Science and Data Center of the Chinese Academy of Sciences (https://www.resdc.cn accessed on 12 December 2020) |
| | | Water network density ($X_6$) | Analysis of water system vector data |
| | | Soil type ($X_7$) | National Earth System Science Data Center (http://geodata.nnu.edu.cn accessed on 12 December 2020) |
| | Socio-economic | GDP per capita ($X_8$) | - |
| | | Population density ($X_9$) | - |
| | | Road density ($X_{10}$) | - |
| | | Nature reserve density ($X_{11}$) | Geographic Information Database of Specimen Resource Sharing Platform in China Natural Reserve (http://www.papc.cn accessed on 12 December 2020) |
| | | Land use type ($X_{12}$) | Resource and Environmental Science and Data Center of the Chinese Academy of Sciences (https://www.resdc.cn accessed on 12 December 2020) |

In this study, land use type data from the study area was used to determine and describe the response of different habitat types to threat sources and to obtain habitat distribution characteristics. The InVEST model suggests that the higher the habitat quality, the higher the corresponding biodiversity, and the more stable the ecosystem. The equation for the habitat quality index was as follows:

$$Q_{xj} = H_j \left[ 1 - \left( \frac{D_{xj}^z}{D_{xj}^z + k^z} \right) \right] \tag{1}$$

where $Q_{xj}$ is the habitat quality index of grid $x$ in land use type $j$, ranging from 0 to 1. $H_j$ is the habitat suitability of land use type $j$. $D_{xj}$ represents the degree of habitat degradation in grid $x$ of land use type $j$. $k$ is the half-saturation coefficient. $z$ is the default parameter of the model, which is a normalized constant, usually taken as 2.5.

According to Equation (1), another very important parameter that affects the calculation results of habitat quality is the degree of habitat degradation ($D_{xj}$). The calculation equation was as follows:

$$D_{xj} = \sum_{r=1}^{R} \sum_{y=1}^{Y_r} \left( \frac{w_r}{\sum_{r=1}^{R} w_r} \right) r_y i_{rxy} \beta_x S_{jr} \tag{2}$$

where $y$ represents all grids of stress factor $r$. $R$ represents all degradation sources. $Y_r$ represents a set of grids of the stress factor $r$. $w_r$ is the weight of the normalized stress factor $r$, ranging from 0 to 1. $r_y$ is the value of the stress factor $r$ in grid $y$. $i_{rxy}$ is the reachability level of the grid $x$. $\beta_x$ is the approachable level of the grid $x$. $S_{jr}$ is the sensitivity of land use type $j$ to stress factor $r$. Among them, the calculation equation of the $i_{rxy}$ was as follows:

$$\text{Linear decay function}: i_{rxy} = 1 - \left( \frac{d_{xy}}{d_{rmax}} \right) \tag{3}$$

$$\text{Exponential decay function}: i_{rxy} = exp \left[ - \left( \frac{2.99}{d_{rmax}} \right) d_{xy} \right] \tag{4}$$

where $d_{rmax}$ is the maximum influence distance of habitat stress factor $r$. $d_{xy}$ is the linear distance between grid $x$ and $y$.

Based on reference to the existing literature [6,11], the InVEST model user manual, and the current situation of habitats in the BAYRAP, this study has set parameters such as the maximum impact distance and weight of each threat factor, and the sensitivity of habitat types to threat factors (Tables 2 and 3). Its value range was 0~1. Land use/land cover data from 1990 to 2020 were sourced from the Resource and Environmental Science and Data Center of the Chinese Academy of Sciences (https://www.resdc.cn/ accessed on 24 December 2020) (Figure 2).

**Table 2.** Weight and influence distance of threat factors.

| Threat Factor $r$ | Maximum Impact Distance $d_{max}$ | Weight $w$ | Decay Function $i$ |
|---|---|---|---|
| Urban land | 10 | 0.9 | Exponential decay |
| Rural residential area | 6 | 0.6 | Exponential decay |
| Other construction land | 5 | 0.5 | Exponential decay |
| Paddy land | 1 | 0.3 | Linear Decay |
| Dry land | 1 | 0.3 | Linear Decay |

**Table 3.** Sensitivity of habitat types to threat factors.

| Land Types | Habitat Suitability | Urban Land | Rural Residential Area | Other Construction Land | Paddy Land | Dry Land |
|---|---|---|---|---|---|---|
| Paddy land | 0.3 | 0.5 | 0.6 | 0.5 | 0 | 1 |
| Dry land | 0.3 | 0.5 | 0.6 | 0.5 | 1 | 0 |
| Forest land | 1 | 0.7 | 0.7 | 0.7 | 0.8 | 0.7 |
| Shrub wood | 0.9 | 0.6 | 0.5 | 0.6 | 0.7 | 0.6 |
| Sparse wood | 0.7 | 0.8 | 0.7 | 0.6 | 0.7 | 0.7 |
| Other forest land | 0.5 | 0.6 | 0.7 | 0.6 | 0.4 | 0.5 |
| High coverage grassland | 0.8 | 0.6 | 0.7 | 0.4 | 0.6 | 0.7 |
| Moderate coverage grassland | 0.6 | 0.6 | 0.6 | 0.5 | 0.5 | 0.5 |
| Low coverage grassland | 0.5 | 0.6 | 0.5 | 0.5 | 0.4 | 0.5 |
| River and canals | 0.9 | 0.5 | 0.4 | 0.4 | 0.4 | 0.4 |
| Lakes | 1 | 0.7 | 0.6 | 0.5 | 0.6 | 0.7 |
| Reservoirs and ponds | 0.9 | 0.6 | 0.6 | 0.4 | 0.5 | 0.6 |
| Mudflat | 0.8 | 0.7 | 0.8 | 0.6 | 0.6 | 0.4 |
| Urban land | 0 | 0 | 0 | 0 | 0 | 0 |
| Rural residential area | 0 | 0 | 0 | 0 | 0 | 0 |
| Other construction land | 0 | 0 | 0 | 0 | 0 | 0 |
| Bare land | 0.1 | 0.2 | 0.1 | 0.1 | 0 | 0 |

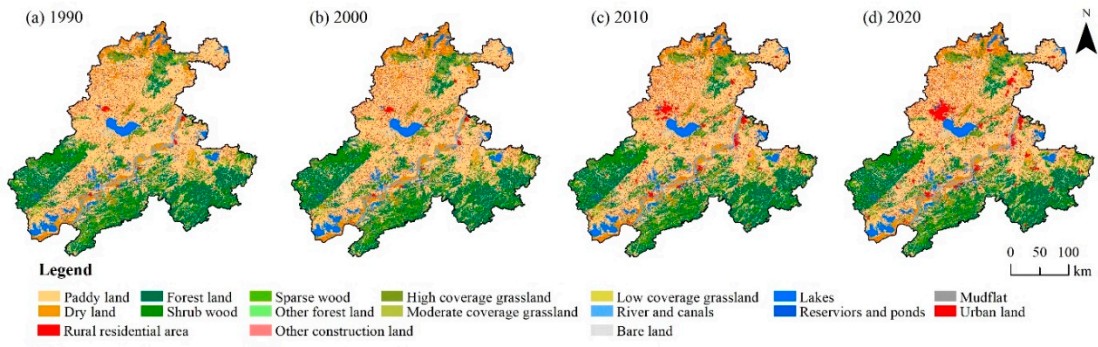

**Figure 2.** Land use distribution and changes in the basin along the Yangtze River in Anhui Province.

To deeply understand the temporal and spatial changes in habitat quality in the BAYRAP, this study divided the evaluation results of habitat quality into five grades based on the criteria of lowest (0~0.2), low (0.2~0.4), middle (0.4~0.6), high (0.6~0.8), and highest (0.8~1.0).

### 2.3.2. Hot and Cold Spot Analysis

Hot and cold spot analysis is a widely used geospatial statistical analysis method. Its application in habitat quality mainly reflected the spatial aggregation and differentiation characteristics of quality [31,37]. The study area was divided into a 1 km $\times$ 1 km grid, which was used as the minimum unit for calculating the spatial aggregation of habitat quality. This study used the local *Moran's I* index to calculate the spatial clustering characteristics of habitat quality. The calculation formula was as follows:

$$I = \frac{n\sum_{i=1}^{n}\sum_{j=1}^{n}w_{ij}(x_i - \bar{x})(x_j - \bar{x})}{\sum_{i=1}^{n}(x_i - \bar{x})^2\left(n\sum_{i=1}^{n}\sum_{j=1}^{n}w_{ij}\right)} \tag{5}$$

$$Z(G_i^*) = \frac{\sum_{j=1}^{n}w_{ij}x_j - \bar{x}\sum_{j=1}^{n}w_{ij}}{\sqrt[2]{\frac{n\sum_{j=1}^{n}w_{ij}^2 - \left(\sum_{j=1}^{n}w_{ij}\right)^2}{n-1}}} \tag{6}$$

where $I$ represent *Moran's I*, $Z(G_i^*)$ represents the cold and hot spot index, $n$ represents the number of grid cells in the study area, and $x_i$ and $x_j$ represent the observed values of spatial unit $i$ and $j$, respectively. $\bar{x}$ is the average value of the spatial unit, and $x_{ij}$ is the weight matrix of the spatial units $i$ and $j$. The *Moran's I* have a value of $[-1, 1]$. The significance of its statistics was tested using the Monte Carlo simulation method, with a simulation frequency of 999. The testing equation was as follows:

$$z_I = \frac{I - E(I)}{S(I)} \tag{7}$$

$$S(I) = \sqrt{var(I)} \tag{8}$$

where $E(I)$ is the expected value of the autocorrelation of the observed variable. $var(I)$ and $S(I)$ represent variance and standard deviation, respectively. $z_I$ is a multiple of the standard deviation. At a given significance level, when $z(I) > 0$, it indicates a significant positive correlation between elements. The value of $I$ is close to 1, it indicates the presence of spatial aggregation in spatial units with small spatial differences and similar properties. When $z(I) < 0$, it indicates a significant negative correlation between elements. The closer the $I$ value is to $-1$, the greater the spatial differences between spatial units. $z(I) = 0$ indicates that the spatial autocorrelation of elements is not significant.

### 2.3.3. Geographical Detector

A geographical Detector is a method that reveals the driving force behind elements by detecting their spatial layered heterogeneity, which refers to the phenomenon that the sum of intralayer variances is less than the total variance between layers. The magnitude of the heterogeneity is measured by the $q$ value of the geographical detector [38]. This method was mainly applied to the identification of influencing factors and research on the mechanism of spatial differentiation [39]. The geographical detector included four modules: factor detector, risk detector, interaction detector, and ecological detector. This study used geographical detectors to measure the impact of different factors on the spatiotemporal variability of habitat quality, and the formula was as follows:

$$q = 1 - \frac{\sum_{h}^{L}N_h\sigma_h^2}{N\sigma^2} \tag{9}$$

where $L$ refers to the number of influence factors, and $N_h$ and $N$ refer to the number of units within layer $h$ and the entire study area. $\sigma_h^2$ and $\sigma^2$ are the variances of layer $h$ and the entire study area, respectively. $q$ is the degree of influence of the independent variable on

the spatiotemporal distribution of the dependent variable. $q \in [0, 1]$. The closer the value of $q$ is to 1, the stronger the explanatory power of the independent variable on the dependent variable. Based on the value of $q$, the driving factors of habitat quality could be identified.

Interaction factor detection is the greatest advantage of geographical detectors compared to other statistical methods. It can detect the interaction between two variables and determine the direction and manner of interaction between two factors by comparing the magnitude of single-factor $q$ values and double-factor $q$ values. The judgment basis for interaction was shown in Table 4.

**Table 4.** Two-factor interaction types of geographical detectors.

| Judgment Basis | Interaction |
|---|---|
| $q(X_1 \cap X_2) < [q(X_1), q(X_2)]$ | Nonlinear decay |
| $min[q(X_1), q(X_2)] < q(X_1 \cap X_2) < max[q(X_1), q(X_2)]$ | Single-factor nonlinear decay |
| $q(X_1 \cap X_2) > max[q(X_1), q(X_2)]$ | Double-factor boost |
| $q(X_1 \cap X_2) = q(X_1) + q(X_2)$ | independence |
| $q(X_1 \cap X_2) > q(X_1) + q(X_2)$ | Nonlinear boost |

## 3. Results

### 3.1. Land Use Change

According to the land use distribution map of the BAYRAP in 1990, 2000, 2010, and 2020 (Figure 2) and the area statistics of various types of land after consolidation (Figure 3), the area of paddy land showed a slow decline trend from 34,463.29 km$^2$ in 1990 to 32,043.08 km$^2$ in 2020, with a cumulative decrease of 7.02%. The area of dry land, forest, grassland, and water area has remained basically stable over the past 30 years, with an increase or decrease of around 1% to 2%. Among all land types, the largest increase and decrease in construction land has occurred, with the area increasing from 390.18 km$^2$ in 1990 to 1616.34 km$^2$ in 2020, an increase of 314.26% in the past 30 years, which indicated the continuous acceleration of the land development process in the study area in past 30 years.

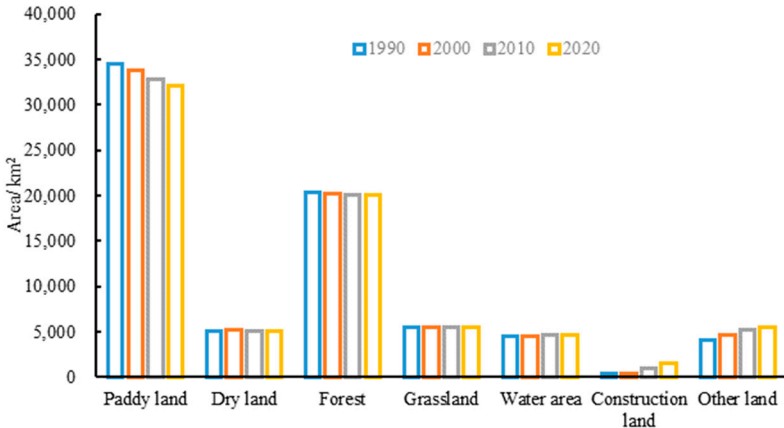

**Figure 3.** Area of each land use type in BAYRAP from 1990 to 2020.

Through spatial superposition of land use type maps of the BAYRAP in different years, the land use transfer matrices of the basin from 1990 to 2000, 2000 to 2010, and 2010 to 2020 were obtained (Table 5). The results showed that there were significant differences in the main directions of land type transfer in the study area in different periods. Among them, the area of other land and dry land in the study area increased the most from 1990 to 2000, with 587.10 km$^2$ and 205.94 km$^2$ respectively. The increase in dry land area was mainly transferred from grassland, forest, and paddy land. During the same period, the largest areas of paddy land and grassland were transferred out, with 706.54 km$^2$ and 214.10 km$^2$, respectively. Paddy land was mainly transferred out for other land use, construction land, dry land, and forest, while grassland was mainly transferred out for dry land and forest.

**Table 5.** Land use transfer matrix in the Yangtze River Basin in Anhui Province (Unit: km$^2$).

| Year | Land Use Type | Grassland | Construction Land | Dry Land | Paddy Land | Water Area | Forest | Other Land |
|---|---|---|---|---|---|---|---|---|
| | Grassland | 5384.96 | 0.18 | 82.91 | 6.16 | 1.96 | 41.42 | 4.42 |
| | Construction land | 0.01 | 389.39 | 0.13 | 0.58 | 0.03 | 0.03 | 0.02 |
| | Dry land | 1.50 | 13.72 | 5020.58 | 3.36 | 0.72 | 25.55 | 46.55 |
| 1990–2000 | Paddy land | 3.34 | 57.58 | 43.89 | 33,756.70 | 34.47 | 41.35 | 525.92 |
| | Water area | 0.64 | 0.28 | 5.50 | 5.02 | 4503.76 | 0.11 | 1.03 |
| | Forest | 101.11 | 1.10 | 72.64 | 29.49 | 0.60 | 20,097.90 | 9.15 |
| | Other land | 0.03 | 2.29 | 0.87 | 9.65 | 5.89 | 0.09 | 4014.12 |
| | Grassland | 5413.12 | 13.01 | 1.53 | 13.23 | 1.83 | 26.88 | 21.95 |
| | Construction land | 0.02 | 462.75 | 0.21 | 1.04 | 0.16 | 0.19 | 0.17 |
| | Dry land | 1.54 | 58.79 | 5076.01 | 12.81 | 5.06 | 4.42 | 67.79 |
| 2000–2010 | Paddy land | 12.69 | 447.48 | 13.43 | 32,578.14 | 119.33 | 77.61 | 562.10 |
| | Water area | 1.70 | 6.25 | 2.42 | 32.63 | 4480.89 | 1.73 | 21.77 |
| | Forest | 22.09 | 29.32 | 6.40 | 78.35 | 1.76 | 20,013.90 | 54.35 |
| | Other land | 0.59 | 17.67 | 6.94 | 56.61 | 10.93 | 1.92 | 4506.54 |
| | Grassland | 5313.61 | 3.72 | 3.48 | 33.05 | 5.04 | 72.70 | 20.09 |
| | Construction land | 1.66 | 1020.73 | 1.46 | 4.91 | 0.89 | 1.36 | 4.29 |
| | Dry land | 4.36 | 35.48 | 4960.24 | 22.52 | 7.50 | 13.61 | 63.07 |
| 2010–2020 | Paddy land | 30.21 | 367.14 | 27.85 | 31,612.94 | 47.94 | 218.57 | 467.63 |
| | Water area | 4.22 | 8.19 | 6.33 | 34.06 | 4533.39 | 5.46 | 27.10 |
| | Forest | 80.01 | 19.13 | 14.70 | 208.76 | 5.01 | 19,751.40 | 45.00 |
| | Other land | 4.91 | 161.94 | 28.24 | 125.99 | 22.45 | 8.20 | 4882.89 |

From 2000 to 2010, other land and construction land in the study area increased the most, with 728.14 km$^2$ and 572.53 km$^2$, respectively. Among them, construction land was mainly transferred from paddy land, with an area transferred in of 447.48 km$^2$, accounting for 78.16% of the area transferred in, followed by dry land and forest, with an area transferred in of 58.79 km$^2$ and 29.32 km$^2$, respectively. Among the land types transferred out, paddy land had the largest transferred-out area, reaching 1232.63 km$^2$. The main transferred-out direction was other land and construction land, with an area of 562.10 km$^2$ and 447.48 km$^2$, respectively, accounting for 81.90% of the total transferred-out area. This indicated that the industrial direction of the study area was accelerating the transformation from agriculture to industrial and urban construction during this period. The situation of land type transfer in and out in 2010–2020 was similar to that in 2000–2010, but the difference was that the area of construction land transfer in 2010–2020 increased significantly, with an increase from 447.48 km$^2$ in the previous period to 595.60 km$^2$. It indicated that the urbanization process continued to accelerate during this period, with rapid changes in land use structure. Among the land types transferred out, paddy land was still the largest type of transferred out, with an area of 1159.34 km$^2$. Its main transfer direction was other land and construction land, with an area of 467.63 km$^2$ and 367.14 km$^2$, respectively. In addition, the process of transferring out of forest and other land during this period also significantly accelerated, with the transfer out areas of 372.62 km$^2$ and 351.74 km$^2$, respectively. The main transfer out directions of the forest was paddy land and grassland, while the main transfer out directions of other land was construction land and paddy land.

### 3.2. Analysis of Spatiotemporal Evolution of Habitat Quality

3.2.1. Characteristics of Spatiotemporal Differentiation of Habitat Quality

According to the results of the InVEST model, the habitat quality index layer of the BAYRAP from 1990 to 2020 was obtained. According to the principle of habitat quality grading, the habitat quality index was divided into five grades, and the habitat quality grading map of the study area was obtained (Figure 4). The results showed that there was little change in the range of low and highest grades of habitat quality between 1990 and 2020. The regions with obvious changes in habitat quality grades were mainly concentrated along the Yangtze River and in the northern part of the Chaohu Lake. These regions were

urban concentrated areas in the BAYRAP, with increasing urban development activities and poor ecosystem stability. The western and southern regions of the study area had high or highest habitat quality grades, with the western area belonging to the Dabie Mountains and the southern area belonging to the Southern Anhui Mountain Area. These regions were dominated by forest ecosystems with relatively complex structures, high levels of biodiversity, and fewer human development activities, which resulted in high habitat anti-interference and stability.

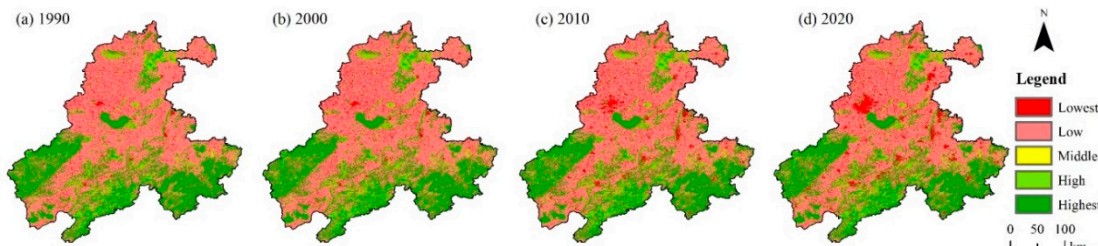

**Figure 4.** Habitat quality grades changed in the BAYRAP from 1990 to 2020.

According to the statistics of the proportion of areas with different grades of habitat quality (Figure 5), the area with a low grade of habitat quality in the study area showed a decreasing trend year by year from 1990 to 2020, while the proportion of areas with the lowest grade showed an increasing trend year by year, with an increase from 4.85% in 1990 to 8.47% in 2020. Overall, the proportion of areas with high and highest grades of habitat quality during various periods has shown a dynamic and stable trend. Based on Figure 4, it could be seen that areas with high habitat quality in the basin were concentrated in the western and southern mountainous areas. The ecosystem structure in these areas was relatively stable. Therefore, the habitat quality has remained stable for 30 years, which enhanced the anti-interference ability of the ecosystem in the BAYRAP.

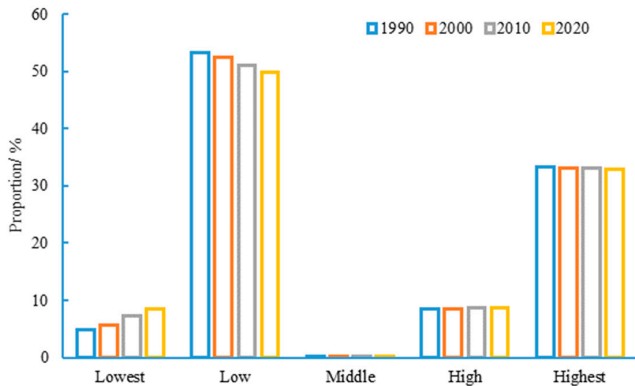

**Figure 5.** Area proportion of different habitat quality grades in the BAYRAP from 1990 to 2020.

### 3.2.2. Analysis of Hot and Cold Spots of Habitat Quality Degradation

To more comprehensively display the spatiotemporal differentiation characteristics of habitat quality degradation in the basin, this study used the hotspot analysis module of ArcGIS software to reveal the spatial aggregation characteristics of habitat quality degradation from 1990 to 2020 (Figure 6). Before the hot and cold spot analysis, global autocorrelation (*Moran's I*) tests were used to calculate the parameter values of habitat degradation in four periods. The results showed that the *Moran's I* of habitat quality degradation in 1990, 2000, 2010, and 2020 were 0.73, 0.82, 0.79, and 0.85, respectively, with $p$ values less than 0.01, $Z$ scores of 69, 92, 87, and 93, all greater than 2.58, which indicated that the spatial distribution of habitat degradation had a uniform and significant spatial correlation, and the hot and cold spot analysis had statistical significance.

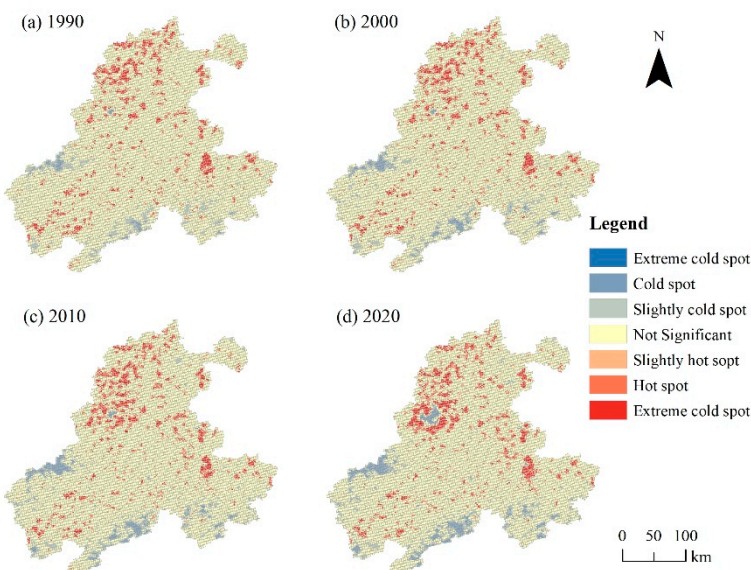

**Figure 6.** Distribution map of cold and hot spots of habitat quality degradation in the BAYRAP from 1990 to 2020.

The results of hot and cold spot analysis indicated that the degree of habitat degradation was a hot spot in the northern area of the basin, while the cold spot was mainly concentrated in the western and southern areas of the basin. As time went on, the hot spots of habitat degradation in the northern area of the basin had shown a continuous trend of diffusion. The northern area of the basin is Hefei City, the capital of Anhui Province. In the past 30 years, the rapid development of urbanization, changes in land use patterns, and the gradual concentration of population in the area around the Chaohu Lake had led to an increasing degree of regional habitat degradation. The pressure on the ecological environment in the area had gradually become prominent, which had to some extent damaged the integrity of the regional ecosystem. After overlaying the habitat quality evaluation and the habitat quality degradation degree layer, the areas with high habitat quality in the study area highly overlap with the cold spots of habitat degradation degree, i.e., the hot spots of habitat degradation degree and the habitat quality was generally in the opposite trend. It indicated that concentrated contiguous mountainous areas with high habitat quality had a higher level of ecosystem stability. As ecological reserves, these areas had a high aggregation trend, which would contribute to the integrity and stability of the ecological protective screen in the basin.

### 3.3. Analysis of Driving Forces for Spatial Differentiation of Habitat Quality

#### 3.3.1. Key Driving Factors

The factor detection results based on geographical detectors showed that all driving factors had passed the significance test at the 0.01 level. It could be seen from the single-factor detection results (Table 6) that there were differences in the interpretation of various driving factors on the habitat quality in the study area. The specific explanatory power levels ranged from large to small: land use > NDVI > slope > population density > water network density > GDP per capita > soil type > altitude > road density > natural reserve density > annual temperature > annual precipitation. Overall, except for land use type, the effect of the natural environment and socio-economic factors on the driving forces of habitat quality in the study area was relatively balanced.

**Table 6.** The *q* value of influencing factors of spatial differentiation characteristics of habitat quality.

| Influencing Factors | Land Use Type | NDVI | Slope | Population Density | Water Network Density | GDP per Capita | Soil Type | Altitude | Road Density | Nature Reserve Density | Annual Temperature | Annual Precipitation |
|---|---|---|---|---|---|---|---|---|---|---|---|---|
| *q* value | 0.638 | 0.382 | 0.305 | 0.217 | 0.139 | 0.103 | 0.087 | 0.043 | 0.036 | 0.033 | 0.015 | 0.010 |

The explanation of a phenomenon by a single factor was often less powerful than the interaction of various factors. To explore the influence of the interaction between factors on habitat quality, this study explored the interaction of habitat quality at the basin scale (Figure 7). The results showed that the interaction between NDVI and soil type was the strongest among natural environmental factors, with a *q* value of 0.682, which indicated that these two factors had the greatest comprehensive influence on habitat quality among natural environmental factors. Among the socio-economic factors, the interaction between GDP per capita and land use type was the strongest, with a *q* value of 0.747. These two factors were also the strongest interacting factors among all 12 factors. The combined effect of the above two factors was the main driving force for the spatiotemporal differentiation of habitat quality in the study area.

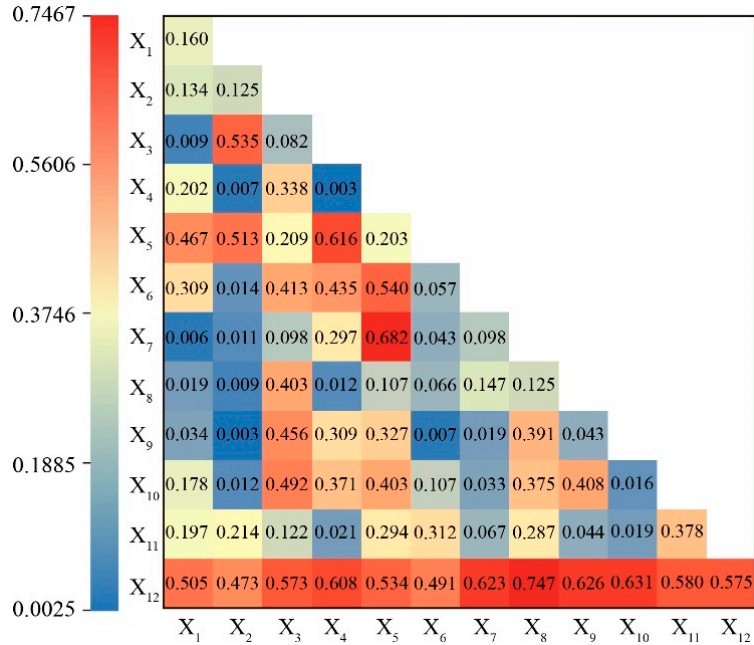

**Figure 7.** Detection results of habitat quality interaction.

### 3.3.2. Analysis of Driving Mechanism

Both single-factor and interactive factor detection results showed that land use type was the most important factor in the evolution of habitat quality in the study area. Therefore, to more intuitively display the influence of land use on the habitat quality in the BAYRAP, this study conducted a visual display of land use conversion in the study area from 1990 to 2020 (Figure 8). The results showed that in the past 30 years, the retention rate of forest and paddy land was the highest among the land use types in the study area. As ecological land, forests promoted the maintenance and improvement of habitat quality in some areas of the study area, while paddy land, as an artificial wetland, had a weak negative influence on habitat quality. As the main threat source in the study area, construction land was mainly transferred from paddy land, and rapid urban expansion has led to the requisition of agricultural land around the city for roads, industrial parks, and urban housing. The increase in urban construction land and the decline in the number of grasslands and woodlands have led to a decline in the quality of the habitat in the BAYRAP.

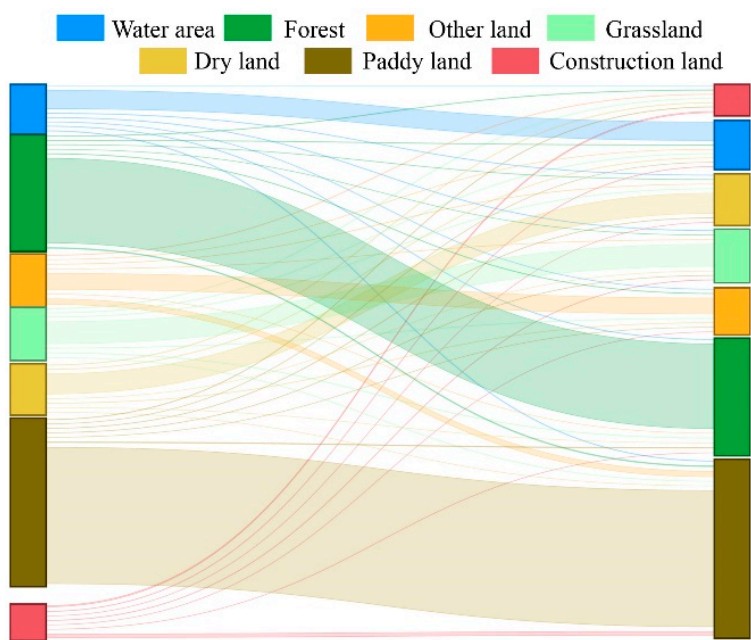

**Figure 8.** Sankey map of land use conversion from 1990 to 2020.

In addition, there was a significant interaction between land use and other factors, which indicated that human activities had a profound effect on regional ecosystem types, determining the evolution of the basin ecosystem patterns, and thus restricting the improvement of habitat quality in the BAYRAP.

## 4. Discussion and Conclusions

### 4.1. Discussion

The BAYRAP is a transitional zone from the hilly and mountainous areas in the middle and lower reaches of the Yangtze River to the plain (the Taihu Lake Plain). The terrain in the basin is low in the northeast and high in the southwest. The north-south changes in terrain and the north-south extension of the Yangtze River water system promote the regional differences in vegetation, soil, and land use types in the BAYRAP. The terrain along the Yangtze River and its northern area is relatively flat, with dozens of lakes and wetlands of different sizes distributed along the river [27,40]. They are important ecological sources in the region and are also important habitats and wintering grounds for overwintering waterbirds in China [25,41]. With the rapid development of urbanization in the past 30 years and the rapid development of economics, large areas of paddy land and natural land along the Yangtze River and its northern regions have been developed for transportation and construction purposes. The regional ecological landscape has shown a fragmented trend. The area of forest land and grassland has continued to decrease, the living space of rare and endangered species has been gradually compressed [42], and the declining trend of habitat quality in the basin has gradually appeared. The simulation results using the InVEST model in this study also showed that the habitat quality along the Yangtze River and its northern areas has been deteriorating continuously in the past 30 years. The areas to the south and west of the Yangtze River belong to mountainous areas, and the level of human development and utilization has been relatively low in the past 30 years. As an important ecological protective screen in the middle and lower reaches of the Yangtze River, the ecological environment protection of the mountainous areas in southern Anhui and the Dabie Mountains has always been the focus of comprehensive ecological environment protection in Anhui Province. Therefore, over the past 30 years, the ecosystem structure of the two regions has remained intact, the ecological environment in some regions has been continuously optimized, and the level of biodiversity has remained stable, which promoted the stability and improvement of the habitat quality in these regions.

As is well known, structural changes in land use are one of the main reasons for the deterioration of the regional ecological environment [43]. Therefore, before conducting ecosystem assessment and habitat quality assessment, it is necessary to analyze and discuss the land use situation in the basin. The results of this study showed that the area of dry land, forest, grassland, and water area has remained relatively stable in the past 30 years. These land types correspond to fields such as agriculture, forestry, and natural ecology, respectively. It indirectly showed that low-intensity or ecological land in the basin has maintained a certain scale in the past 30 years, which supported the stability of habitat quality in some areas of the basin. In the rapidly developing Yangtze River Delta region, the stable land use structure of the BAYRAP seems to be out of place. The reason for this was mainly related to two factors. Firstly, the overall stability of regional land use structure and intensity. Previous studies have shown that unreasonable land development activities within the basin are the main cause of habitat loss for species [44,45]. Although the BAYRAP is located in the Yangtze River Delta Economic Belt, it is far from economically developed cities such as Shanghai and Nanjing. The region was only included in the Yangtze River Delta Economic Zone from the perspective of Chinese national policies in the past ten years. Therefore, the land use structure has shown an overall stable trend in the past 30 years. Especially at the end of the last century and the beginning of this century, due to the abundant water resources, the region became an important agricultural planting area [46], with slow urban and industrial development activities, which to some extent maintained the stability of the ecosystem. Secondly, the terrain within the basin is diverse and the water system is well-developed. As a transitional zone from mountainous areas in the middle and lower reaches of the Yangtze River to coastal plains, the BAYRAP has diverse terrain. The northern and southern parts of the basin are both mountainous areas, which form a nearly trumpet-shaped plain along the river. A large number of rivers converge here and flow into the Yangtze River, and the population in this area is highly concentrated in large and medium-sized cities such as Anqing, Tongling, and Wuhu. The population growth rate has been moderate in the basin, and the evidence, of the growth of the area of construction land in this study in the past 30 years was mainly concentrated in the vicinity of cities and towns, was reflected in the study. However, as the BAYRAP gradually integrates into the Yangtze River Delta Economic Belt, the speed of economic development and population size of the basin will grow rapidly. Therefore, the changes in land use structure in the basin may accelerate in the future, and its impact on habitat quality will deserve further attention.

This study conducted a basin-scale habitat quality assessment using the InVEST model, which revealed the spatiotemporal evolution of habitat quality in a basin over the past 30 years. As one of the methods for visual accounting of ecological value, the InVEST model has been widely used in basin ecosystem assessment in recent years [22,31], which effectively revealed changes in the quality and value of basin ecosystems under different land cover scenarios. As an important component of the Yangtze River Delta economic Belt and the core area of the ecological protective screen in the middle and lower reaches of the Yangtze River, the BAYRAP has dual pressures of economic and social development and ecological environmental protection. Based on past research [47–49], it is found that relevant research on the BAYRAP mainly focused on the landscape or regional economic and social development aspect, such as the spatiotemporal changes of wetland habitats in the basin of Shengjin Lake, Caizi Lake, and Anqing riverside lake wetland groups, as well as the spatiotemporal differentiation characteristics of economic and population development in the riverside urban belt. Although these studies have revealed changes in the human–land relationship and the evolution of wetland species habitat suitability levels in the BAYRAP from their respective perspectives, they could not reveal the internal relationship between local ecological environment changes and regional human activities, and could not analyze the ecological environmental effects of human activities from a macro level. This study was based on the InVEST model, using the ecological sensitivity characteristics covered by land use types to penetrate the internal relationship between

human activities and ecosystem quality, to demonstrate the impact mechanism of human activity changes on ecosystem and ecological environment quality in the basin.

There are more than ten national and provincial nature reserves, as well as internationally important wetlands and wetland parks distributed in the BAYRAP. The value of regional ecological services is enormous, which supports human well-being and the improvement of biodiversity in the region [50,51]. In response to the continuous decline in habitat quality in some regions of the BAYRAP, human development activities should be appropriately restricted while taking into account economic and social development and ecological environment quality. Firstly, development activities should be restricted to non-traditional populations or urban agglomeration areas along the river to protect the ecological environment along the river and the water quality. Secondly, key cities along the river should focus on connotation development, reduce the encroachment on surrounding farmland, wetlands, and ecological sources, and promote the sustainable development of urbanization. In mountainous areas in the south and west, as well as along rivers, lakes, and wetlands, development and agricultural activities should be reduced to reduce the impact of human activities on water bodies, animals, and plants. In particular, in mountainous areas, forest land conservation should be conducted effectively, and natural heritage protection should be carried out in strict accordance with the ecological protection red line requirements. The ecological protective screen function of the Dabie Mountains and the mountainous areas in southern Anhui in the BAYRAP should be fully strengthened.

### 4.2. Conclusions

The conclusions of this study were as follows: (1) The area of construction land in the BAYRAP increased by 314.26% from 1990 to 2020, while the area of paddy land decreased slowly, with a cumulative decrease of 7.02%. There were differences in the dominant transfer types of land use in different periods. The main performance during 1990–2000 was the transfer in of dry land and other land areas. Between 2000 and 2010, other land and construction land had the largest area of transferred in, while paddy land had the largest area of transferred out. Between 2010 and 2020, the largest amount of construction land was transferred, which indicated that the urban process continued to accelerate during this period. Paddy land had the largest area of transferred out and was mainly transferred out for other land and construction land. (2) The results of habitat quality analysis showed that the area proportion of the lowest grade of habitat quality in the study area increased year by year from 1990 to 2020, with an increase from 4.85% in 1990 to 8.47% in 2020. The hot spots of habitat quality degradation were mainly concentrated in Hefei City and its surrounding areas, while the cold spots were mainly concentrated in the Dabie Mountains and the Southern Anhui Mountains in the basin. These regions had good ecosystem stability, high biodiversity levels, and stable habitat quality. (3) The single-factor detection results showed that land use type was the most important driving factor affecting the habitat quality in the study area. The interaction detection results showed that the interaction between GDP per capita and land use type was the strongest, with a $q$ value of 0.747, which indicated that the joint action of the above two factors was driving the evolution of habitat quality in the study area.

**Author Contributions:** Conceptualization, Y.C. and C.W.; Data curation, Y.S., H.D., X.W. and R.L.; Formal analysis, Y.C.; Investigation, X.W.; Resources, Q.S.; Software, Y.W. and Z.C.; Visualization, R.L., Q.S., Y.W. and Z.C.; Writing—original draft, Y.C.; Writing—review & editing, C.W. All authors have read and agreed to the published version of the manuscript.

**Funding:** This research was funded by the National Natural Science Foundation of China [Grant Nos. 32201346, 42201281, 32271662, 42101105], Natural Science Foundation of Anhui Province, China [No. 2208085QD102], the Introducing and Stabilizing Talent Project of Anhui Agricultural University [No. yj2020-47, No. rc402201], and Prosperity and Development Philosophy and Social Science Fund Project of Anhui Agricultural University [No. 2020zs06zd]. The APC was funded by grant number [32201346].

**Institutional Review Board Statement:** Not applicable.

**Informed Consent Statement:** Not applicable.

**Data Availability Statement:** Data is unavailable due to privacy or ethical restrictions.

**Acknowledgments:** Thanks to the reviewers for their contribution to improving the quality of this manuscript.

**Conflicts of Interest:** No conflict of interest exists in the submission of this manuscript, and the manuscript is approved by all authors for publication. On behalf of my co-authors, I would like to state that the work described was an original study that has not been published before and is not under consideration for publication elsewhere, in whole or in part.

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
