# Peer review of "Study on Spatiotemporal Evolution and Driving Forces of Habitat Quality in the Basin along the Yangtze River in Anhui Province Based on InVEST Model"

_land, doi:10.3390/land12051092_

Round 1

Reviewer 1 Report

The authors present interesting findings based on InVEST model. They found variability in land use change between 1990 and 2020.

This is a very important study that is relevant to the Land Journal. Although I recommend acceptance of the article, several issues should be addressed to enhance its suitability. The discussion must come before the conclusion.

The entire document requires an English speaker to correct obvious mistakes.

The abstract should be restructured.

The authors write “The study results showed that the trend of increase and decrease of construction land and paddy land in the basin from 1990 to 2020 was opposite” Did the construction land decrease or increase?

What did you mean by “the largest amount of construction land was transferred in and the largest amount of paddy land was transferred out in other periods.”

Review the Keyword, e.g., “The basin along the Yangtze River in Anhui Province” is not a keyword.

“After reviewing the research literature, it could be found that” should read, “it was found”

You mentioned several ecological models but decided to use InVEST, why?

The paragraph begins with, “Although habitat quality assessment has been widely used at basin scales,).” And ends with; “in the Yangtze River basin” should be properly cited with references.

You obtained very good results with evidenced land use change. Some sections remained stable while others decreased. Tell us how and why this is so. Interrogate your results with previous findings. Is there any difference between your current study and with previous and why? For instance, why did the area of dry land, forest, grassland, and water remain stable over the past 30 years? What happened?

You have indicated that there was no significant change in the range of low and highest grades of habitat quality between 1990 and 2020. Did you test the significance? At what alpha level did you test that?

The entire article requires editing by a native English speaker

Author Response

Dear Reviewer #1,

  Thank you for your comments on the manuscript. Your constructive comments helped to clarify the concepts of the paper. According to your comments, we made modifications to the manuscript. Our detailed responses to your comments are below.

Comment 1: The discussion must come before the conclusion.

Response: We fully adopted your suggestion. In the revised manuscript, we have adjusted the order of discussions and conclusions.

Comment 2: The entire document requires an English speaker to correct obvious mistakes. The abstract should be restructured. The authors write “The study results showed that the trend of increase and decrease of construction land and paddy land in the basin from 1990 to 2020 was opposite” Did the construction land decrease or increase? What did you mean by “the largest amount of construction land was transferred in and the largest amount of paddy land was transferred out in other periods.” Review the Keyword, e.g., “The basin along the Yangtze River in Anhui Province” is not a keyword.

Response: Thanks for your kind advice. In the revised manuscript, we have made modifications to the Abstract section. Firstly, we have made modifications to expressions that have semantic errors or are prone to misunderstandings. For example, we have added an explanation of the trend of increase and decrease in construction land and paddy land in lines 25 to 26 of the revised manuscript. The specific content is: “The study results showed that the trend of increase and decrease of construction land and paddy land in the basin from 1990 to 2020 was opposite, that is, the area of construction land increased and the area of paddy land decreased”. Secondly, we have also modified “Except for 1990-2000, the largest amount of construction land was transferred in and the largest amount of paddy land was transferred out in other periods” in original version to “During the period from 2000 to 2020, Construction land was mainly transferred in from paddy land, accounting for over 60% of the area of transferred in” in the revised version.

Moreover, we have deleted “The basin along the Yangtze River in Anhui Province” from the keywords, and added “Basin” as the new keyword.

Comment 3: “After reviewing the research literature, it could be found that” should read, “it was found”. You mentioned several ecological models but decided to use InVEST, why?

Response: We have fully adopted your suggestion. In line 63 of the revised manuscript, we have changed the relevant expressions. We deleted the words “After reviewing the research literature, it could be found that”. We have provided detailed supplementary explanations in the Introduction section of the revised manuscript regarding the reason for choosing the INVEST model. The specific content was that “It should be pointed out that among numerous ecological models, the InVEST model is widely used in the quantitative spatial assessment of ecosystem services (Fu et al., 2014), and this model mainly evaluates habitat quality based on the comprehensive impact of various threats on the substrate where the habitat patches are located. Therefore, it has a certain comprehensive advantage in regional habitat quality assessment. However, other models heavily rely on the support of sample points or species distribution data, which would easily lead to bias in evaluation results, and these models weaken the impact of differences in habitat patches or landscape structure types.”

Comment 4: The paragraph begins with, “Although habitat quality assessment has been widely used at basin scales,).” And ends with; “in the Yangtze River basin” should be properly cited with references.

Response: We have fully adopted your suggestion. In lines 99 to 101 of the revised manuscript, we have added three citations. The specific content was that Although habitat quality assessment has been widely used at basin scales (Dong et al., 2022; Wei et al., 2022), there was still a gap in the study of habitat quality in the basin along Yangtze River in Anhui Province (BAYRAP) (Zhong et al., 2022).

The cited literatures were:

Wei, Q., Abudureheman, M., Halike, A., Yao, K., Yao, L., Tang, H., Tuheti, B., 2022. Temporal and spatial variation analysis of habitat quality on the PLUS-InVEST model for Ebinur Lake Basin, China. Ecological Indicators 145.

Dong, J., Zhang, Z., Liu, B., Zhang, X., Zhang, W., Chen, L., 2022. Spatiotemporal variations and driving factors of habitat quality in the loess hilly area of the Yellow River Basin: A case study of Lanzhou City, China. Journal of Arid Land 14, 637-652

Zhong, C., Bei, Y., Gu, H., Zhang, P., 2022. Spatiotemporal Evolution of Ecosystem Services in the Wanhe Watershed Based on Cellular Automata (CA)-Markov and InVEST Models. Sustainability 14.

Comment 5: You obtained very good results with evidenced land use change. Some sections remained stable while others decreased. Tell us how and why this is so. Interrogate your results with previous findings. Is there any difference between your current study and with previous and why? For instance, why did the area of dry land, forest, grassland, and water remain stable over the past 30 years? What happened?

Response: Thanks for your suggestion. We discussed the reasons for land use change in the discussion section of the revised manuscript. The specific content was as follow:

As is well known, structural changes in land use are one of the main reasons for the deterioration of regional ecological environment (Li et al., 2017). Therefore, before conducting ecosystem assessment and habitat quality assessment, it is necessary to analyze and discuss the land use situation in the basin. The results of this study showed that the area of dry land, forest, grassland, and water area has remained relatively stable in the past 30 years. These land types correspond to fields such as agriculture, forestry, and natural ecology, respectively. It indirectly showed that low intensity or ecological land in the basin has maintained a certain scale in the past 30 years, which supported the stability of habitat quality in some areas of the basin. In the rapidly developing Yangtze River Delta region, the stable land use structure of the BAYRAP seems to be out of place. The reason for this was mainly related to two factors. Firstly, the overall stability of regional land use structure and intensity. Previous studies have shown that unreasonable land development activities within basin are the main cause of habitat loss for species (McCoshum and Geber, 2019; Platenberg and Harvey, 2010). Although the BAYRAP is located in the Yangtze River Delta Economic Belt, it is far from economically developed cities such as Shanghai and Nanjing. The region was only included in the Yangtze River Delta Economic Zone from the perspective of Chinese national policies in the past ten years. Therefore, the land use structure has shown an overall stable trend in the past 30 years. Especially at the end of the last century and the beginning of this century, due to the abundant water resources, the region became an important agricultural planting area (Zhao et al., 2022), with slow urban and industrial development activities, which to some extent maintained the stability of the ecosystem. Secondly, the terrain within the basin is diverse and the water system is well-developed. As a transitional zone from mountainous areas in the middle and lower reaches of the Yangtze River to coastal plains, the BAYRAP has diverse terrain. The northern and southern parts of the basin are both mountainous areas, which form a nearly trumpet shaped plain along the river. A large number of rivers converge here and flow into the Yangtze River, and the population in this area is highly concentrated in large and medium-sized cities such as Anqing, Tongling, and Wuhu. The population growth rate has been moderate in the basin, and the evidence, the growth of the area of construction land in this study in the past 30 years was mainly concentrated in the vicinity of cities and towns, was reflected in the study. However, as the BAYRAP gradually integrates into the Yangtze River Delta Economic Belt, the speed of economic development and population size of the basin will grow rapidly. Therefore, the changes in land use structure in the basin may accelerate in the future, and its impact on habitat quality will deserve further attention.

Comment 6: You have indicated that there was no significant change in the range of low and highest grades of habitat quality between 1990 and 2020. Did you test the significance? At what alpha level did you test that?

Response: I'm sorry for the doubts caused to your review due to the imprecise wording. Therefore, in lines 303-305 of the revised manuscript, we have changed the expressions of changes in habitat quality trends. At the same time, we replaced “significant” in the original version with “obvious”. We overlooked the statistical significance of the word “significant” in the original manuscript. The specific revised content was that “The results showed that there was little change in the range of low and highest grades of habitat quality between 1990 and 2020.”

We need to explain that truth that the InVEST model is applied to the habitat quality by simulated the habitat patches of landscape. Especially, the InVEST model belongs to black-box model, so the simulation results are output directly. The data format of the results is raster, so we input the raster data to ArcGIS platform to divide the quality grades. Finally, we obtained the map of the habitat quality grades. Obviously, the process of the quality assessment was not involved the significance. Importantly, the Habitat Quality module in InVEST model needn’t test alpha level.

Reviewer 2 Report

The authors examine an interesting topic (Study on spatiotemporal evolution and driving forces of habitat quality in the basin along the Yangtze River in Anhui Province based on InVEST model?). But the methodology section does not contain the detailed data analysis (correlation) that is necessary for scientific publications. The author used multiple threats, but no every mentioned requirement for each indicator used, the authors must describe the specifics, and the justification for the necessity of select indicator used. Most importantly, the author should mention the implication of the study as separate paragraph.   However, the worthy data in the manuscript deserve publication and thus I give some advice with which the authors could make a new manuscript which could be published for example in this journal.

Author Response

Dear Reviewer #2,

Thank you for your high recognition of this study. To describe our research findings more scientifically and accurately, according to your constructive suggestions, we have partially revised the original manuscript. We are very appreciating your contribution for the quality improvement of our manuscript.

Comment 1: But the methodology section does not contain the detailed data analysis (correlation) that is necessary for scientific publications. The author used multiple threats, but no every mentioned requirement for each indicator used, the authors must describe the specifics, and the justification for the necessity of select indicator used. Most importantly, the author should mention the implication of the study as separate paragraph. However, the worthy data in the manuscript deserve publication and thus I give some advice with which the authors could make a new manuscript which could be published for example in this journal.

Response: Thanks for your kind advice. According to your suggestion, we added a paragraph to describe the reason and specific study implication of each indicator in lines 171 to 181 of revised manuscript. The specific revised content was as follow:

The BAYRAP belongs to a subtropical monsoon climate zone, with significant intra and inter annual changes in temperature and precipitation, which reflected differences in the living environment conditions of species. The basin is a transitional zone from mountainous hills to coastal plains, with significant topographic fluctuations, resulting in significant changes in elevation and slope within the basin. NDVI, as an indicator of vegetation ecological status, reflects the level of vegetation health in the basin. Water network density and soil type reflect the water and soil conditions within the basin. In socio-economic conditions, GDP, population, roads, natural reserve, and land types all bear the breadth and depth of human activities, which reflected the potential effect of human production and life on the regional ecological environment and ecosystem structure. The data sources of the above indicator system were described in Table 1.

Reviewer 3 Report

The manuscript used InVEST model to analyze the spatial and temporal evolution of habitat quality based on remote sensing image data from 1990, 2000, 2010, and 2020 in the basin along the Yangtze River in Anhui Province. It revealed the spatial evolution trend of habitat quality degradation using hot and cold spot analysis methods. The author has done much work; nevertheless, it needs further improvements. The reviewer has listed some specific comments that might help the authors further enhance the manuscript's quality.

1.      The author should add line numbers to the article.

2.      The “where” below the formula should have the top grid and the first letter lowercase.

3.      In the abstract, “was used to discuss,” should be “was used to discuss.”

4.      Carefully check the articles 'a',' an', and 'the,' and some mistakes are made.

5.      This article did not perform calibration and validation work on the InVEST model. How can the author prove the accuracy of the calculation results.

Overall, it was pretty good.

Author Response

Dear Reviewer #3,

  Thank you for your comments on the manuscript. Your constructive comments helped to clarify the concepts of the paper. According to your comments, we made modifications to the manuscript. Our detailed responses to your comments are below.

Comment 1: The author should add line numbers to the article.

Response: Thanks for your suggestion. We need to explain to you that the line number was marked in the original manuscript. Of course, the line number would be existent in the revised version.

Comment 2: The “where” below the formula should have the top grid and the first letter lowercase.

Response: We fully adopted your suggestion. We modified the letter case error and ms format error. Please refer to lines 218 and 238 in the revised version for details.

Comment 3: In the abstract, “was used to discussed,” should be “was used to discuss.”

Response: Thanks for your kind advice. We modified some obvious grammatical error in the revised manuscript. the specific modification in the revised manuscript was in line 23.

Comment 4: Carefully check the articles 'a',' an', and 'the,' and some mistakes are made.

Response: Thanks for your suggestion. We carefully checked the grammatical mistakes about ‘a’, ‘an’ and ‘the’. We tried our best to eliminate the mistakes because of the native language difference. If you have any comments on the grammar of this manuscript, we will carefully listen to your suggestions and make necessary modifications.

Comment 5: This article did not perform calibration and validation work on the InVEST model. How can the author prove the accuracy of the calculation results.

Response: We need to explain that truth that the InVEST model is applied to the habitat quality by simulated the habitat patches of landscape. Especially, the InVEST model belongs to black-box model, so the simulation results are output directly. The data format of the results is raster, so we input the raster data to ArcGIS platform to divide the quality grades. Finally, we obtained the map of the habitat quality grades. Obviously, the process of the quality assessment was not involved the significance. Importantly, the Habitat Quality module in InVEST model needn’t test alpha level.

Round 2

Reviewer 2 Report

The methodology section  still does not contain the detailed information on data analysis like correlation and other analysis used by authors. For each of the analyses used, the authors must describe the specifics, the software used, and the justification for the analysis' necessity.  However, the data in the manuscript deserve publication and thus I give some advice with which the authors could add in the manuscript that could be published for example in this journal.

Author Response

Response to the comments from Reviewer #2

Dear Reviewer #2,

  Thank you for your comments on the manuscript. Your constructive comments helped to clarify the concepts of the paper. According to your comments, we made modifications to the manuscript. Our detailed responses to your comments are below.

Comment 1: The methodology section still does not contain the detailed information on data analysis like correlation and other analysis used by authors. For each of the analyses used, the authors must describe the specifics, the software used, and the justification for the analysis' necessity.  However, the data in the manuscript deserve publication and thus I give some advice with which the authors could add in the manuscript that could be published for example in this journal.

Response: We fully adopted your suggestion. In section 2.3 of the revised manuscript, we have added a description of the analysis method and process. The specific content was as follows:

In this study, land use type data from the study area was used to determine and describe the response of different habitat types to threat sources, and to obtain habitat distribution characteristics. The InVEST model suggests that the higher the habitat quality, the higher the corresponding biodiversity, and the more stable the ecosystem. The equation for habitat quality index was as follow:

                                                      (1)

where  is the habitat quality index of grid  in land use type j, ranging from 0 to 1.  is the habitat suitability of land use type .  represents the degree of habitat degradation in grid  of land use type .  is half-saturation coefficient.  is the default parameter of the model, which is a normalized constant, usually taken as 2.5.

According to Eq. (1), another very important parameter that affects the calculation results of habitat quality is the degree of habitat degradation (). The calculation equation was as follow:

                                      (2)

where  represents all grids of stress factor .  represents all degradation sources.   represents a set of grids of the stress factor .  is the weight of the normalized stress factor , ranging from 0 to 1.  is the value of the stress factor  in grid .  is the reachability level of grid .  is the approachable level of grid .  is the sensitivity of land use type  to stress factor . Among them, the calculation equation of the  was as follow:

Linear decay function:                                 (3)

Exponential decay function:                       (4)

where  is the maximum influence distance of habitat stress factor .  is the linear distance between grid  and .

In addition, we have added a description of the analysis method and process of correlation testing in section 2.3.2 of the revised manuscript. The specific content was as follows:

The significance of its statistics was tested using Monte Carlo simulation method, with a simulation frequency of 999. The testing equation was as follow:

                                                        ï¼ˆ7)

                                                   ï¼ˆ8)

where  is the expected value of the autocorrelation of the observed variable.  and  represent variance and standard deviation, respectively.  is a multiple of the standard deviation. At a given significance level, when , it indicates a significant positive correlation between elements. The value of I is close to 1, it indicates the presence of spatial aggregation in spatial units with small spatial differences and similar properties. When , it indicates a significant negative correlation between elements. The closer the I value is to -1, the greater the spatial differences between spatial units.  indicates that the spatial autocorrelation of elements is not significant.

Please refer to lines 190-213, 235-236 and 243-259 of the revised manuscript for details. We hope that the above efforts can dispel the concerns of the reviewer.
